# BRD4 Inhibition as a Strategy to Prolong the Response to Standard of Care in Estrogen Receptor-Positive Breast Cancer

**DOI:** 10.3390/cancers15164066

**Published:** 2023-08-11

**Authors:** Ahmed M. Elshazly, Melanie M. Sinanian, Victoria Neely, Eesha Chakraborty, Muruj A. Alshehri, Michael K. McGrath, Hisashi Harada, Patricia V. Schoenlein, David A. Gewirtz

**Affiliations:** 1Departments of Pharmacology & Toxicology, Virginia Commonwealth University, Richmond, VA 23298, USA; ahmed_elshazly@pharm.kfs.edu.eg (A.M.E.); melanie.sinanian@vcuhealth.org (M.M.S.); alshehrim2@vcu.edu (M.A.A.); 2Massey Cancer Center, Virginia Commonwealth University, Richmond, VA 23298, USA; neelyvl@vcu.edu (V.N.); chakrabortye@vcu.edu (E.C.); hharada@vcu.edu (H.H.); 3Department of Pharmacology and Toxicology, Faculty of Pharmacy, Kafrelsheikh University, Kafrelsheikh 33516, Egypt; 4Philips Institute for Oral Health Research, School of Dentistry, Virginia Commonwealth University, Richmond, VA 23298, USA; 5C. Kenneth and Dianne Wright Center for Clinical and Translational Research, Virginia Commonwealth University, Richmond, VA 23298, USA; 6Department of Pharmacology, College of Clinical Pharmacy, Imam Abdulrahman Bin Faisal University, Dammam 31441, Saudi Arabia; 7Department of Cellular Biology and Anatomy, MCG Cancer Center, Augusta University, Augusta, GA 30912, USA; mimcgrath@augedu.onmicrosoft.com (M.K.M.); pschoenl@augusta.edu (P.V.S.)

**Keywords:** senescence, ARV-825, ABBV-744, BRD4, p53, c-Myc

## Abstract

**Simple Summary:**

The combination of CDK4/6 inhibitors + fulvestrant or tamoxifen effectively prolongs survival in patients with estrogen receptor-positive (ER^+^) breast cancer. However, in the case of residual and metastatic disease, morbidity and mortality are virtually inevitable. Recently, the targeting of dysregulated epigenetic elements, and particularly BET family proteins, has generated considerable interest in the cancer field. The current study was designed to evaluate the capacity of BET inhibitors ARV-825 and ABBV-744 to improve the response to standard-of-care treatment in ER^+^ breast cancer. ARV-825 was effective when combined with tamoxifen in both p53 wild type and p53 null ER^+^ breast cancer cell lines while ABBV-744 showed effectiveness only in combination with fulvestrant plus palbociclib in p53 wild-type cells. Downregulation of both BRD4 and c-Myc are implicated as being required for the sensitizing effects of ARV-825 while c-Myc may not be involved in the case of ABBV-744.

**Abstract:**

Breast cancer is the most commonly occurring malignancy in women and the second most common cause of cancer-related deaths. ER^+^ breast cancer constitutes approximately 70% of all breast cancer cases. The standard of care for ER^+^ breast cancer involves estrogen antagonists such as tamoxifen or fulvestrant in combination with CDK4/6 inhibitors such as palbociclib. However, these treatments are often not curative, with disease recurrence and metastasis being responsible for patient mortality. Overexpression of the epigenetic regulator, BRD4, has been shown to be a negative prognostic indicator in breast cancer, and BET family inhibitors such as ARV-825 and ABBV-744 have garnered interest for their potential to improve and prolong the response to current therapeutic strategies. The current work examined the potential of utilizing ARV-825 and ABBV-744 to increase the effectiveness of tamoxifen or fulvestrant plus palbociclib. ARV-825 was effective in both p53 wild-type (WT) breast tumor cells and in cells lacking functional p53 either alone or in combination with tamoxifen, while the effectiveness of ABBV-744 was limited to fulvestrant plus palbociclib in p53 WT cells. These differential effects may be related to the capacity to suppress c-Myc, a downstream target of BRD4.

## 1. Introduction

Breast cancer is the most commonly occurring malignancy in women [1,2] and the second most common cause of cancer-related deaths, with approximately 40,000 women dying from breast cancer each year in the U.S [2,3]. Among the various types of breast cancer, estrogen receptor alpha-positive (ER^+^) breast cancer is the most common form, constituting approximately 70% of all breast cancer cases [4,5]. The standard of care for this breast cancer subtype is endocrine therapy in combination with adjuvant therapy, which has reduced relapse and mortality by up to 40% [6]. The clinically available endocrine therapies include selective estrogen receptor modulators (SERMs) such as tamoxifen (TAM), selective estrogen receptor degraders (SERDs), such as fulvestrant [7], and aromatase inhibitors (AIs) such as letrozole [5]. SERDs and AIs are currently utilized in combination with CDK4/6 inhibitors such as palbociclib, ribociclib, and abemaciclib. The combination of fulvestrant with palbociclib was reported to extend progression-free survival in ER^+^ breast cancer patients from 4.6 to 11.2 months [8,9,10,11]; however, escape from the growth-suppressive effects of this combination is quite common, leading to disease recurrence.

There has been growing interest in the potential targeting of dysregulated epigenetic elements, such as “super-enhancers” in cancer treatment [12]. Super-enhancers are clusters of enhancers with unusually high levels of transcription factor binding, which are central to driving elevated oncogenic transcription [13]. The bromodomain and extra terminal (BET) protein family, including BRD2, BRD3, and BRD4, by binding to acetylated lysine residues on histone proteins, interact with super-enhancers and epigenetically regulate the transcription of various genes [14]. BET inhibition has demonstrated efficacy in pre-clinical studies and is being evaluated in various clinical trials for both hematological malignancies and solid tumors [15].

As we have shown in a recent publication [8], the combination of fulvestrant with palbociclib drives ER^+^ breast cancer cells into a state of senescence; however, between days 12 and 18, the cells escape from senescence, recovering self-renewal capacity. Administration of a PROTAC BET inhibitor, ARV-825, extended the growth arrest initiated by fulvestrant plus palbociclib in p53 wild-type MCF-7 cells, p53 mutant T47D cells and Rb-deleted ER^+^ breast cancer cell lines [8]. ARV-825 was further shown to have senolytic activity, promoting apoptosis in the senescent breast tumor cells that were initially arrested by the fulvestrant plus palbociclib treatment [8]. ABBV-744 is a BDII domain-selective BET inhibitor, which is currently being evaluated in clinical trials for relapsed/refractory acute myeloid leukemia (AML) as well as in myelofibrosis (NCT03360006, NCT04454658) and has shown promising results in prostate cancer models [16]. In the current work, we investigated the activity of ABBV-744 as a potential adjuvant therapy in combination with fulvestrant plus palbociclib in vitro. We further investigated the potential utility of combining ABBV-744 or ARV-825 with TAM in ER^+^ breast tumor cell lines.

## 2. Materials and Methods

### 2.1. Antibodies and Reagents

The following primary antibodies were used: BRD4 (#13440, Cell Signaling Technology; Danvers, MA, USA); BRD3 (#A16241, ABclonal; Woburn, MA, USA); BRD2 (#A2277, ABclonal); c-Myc (#5605, Cell Signaling Technology); p53 (#9282, Cell Signaling Technology); β-actin (#4970, Cell Signaling Technology); p21 (#2947, Cell Signaling Technology,); and GAPDH (#2118, Cell Signaling Technology). Horseradish peroxidase (HRP)-conjugated anti-mouse (#7076, Cell Signaling Technology) and anti-rabbit (#7074, Cell Signaling Technologies) secondary antibodies were used.

### 2.2. Cell Lines

MCF-7 cells provided by Dr. Erik Knudsen (Roswell Park Comprehensive Cancer Center; Buffalo, NY, USA) were cultured in DMEM (11995-065, Gibco; Waltham, MA, USA,) supplemented with 10% (*v*/*v*) fetal bovine serum (FBS) (26140, GeminiBio; West Sacremento, CA, USA), and 100 U/mL penicillin G sodium and 100 μg/mL streptomycin sulfate (15140122, Gibco). T47D cells were cultured in RPMI medium (30-2001, ATCC; Manassas, VA, USA) supplemented with 10% (*v*/*v*) FBS (SH30066.03, Thermo Scientific; Waltham, MA, USA) and 100 U/mL penicillin G sodium and 100 μg/mL streptomycin sulfate. MCF-7 p53^−/−^ provided by Dr. Xinbin Chen (Comparative Oncology Laboratory, Schools of Medicine and Veterinary Medicine; University of California, Davis, CA, USA) were cultured in RPMI medium (11875-093, Gibco), supplemented with 10% (*v*/*v*) FBS (26140, GeminiBio) and 100 U/mL penicillin G sodium and 100 μg/mL streptomycin sulfate.

### 2.3. Drug Treatment

Fulvestrant (I4409, Millipore Sigma; St. Louis, MO, USA), palbociclib (P-7788, LC Laboratories; Woburn, MA, USA), ABBV-744 (A-1627416, AbbVie; Hong Kong, China), TAM (121893, MedchemExpress; Monmouth Junction, NJ, USA), bafilomycin A1 (196000, Millipore Sigma), and ARV-825 (HY-16954, MedChemExpress) were dissolved in DMSO. Cells were replenished with media or drug-treated media every two days. For fulvestrant and palbociclib studies, cells were exposed to fulvestrant (100 nM) plus palbociclib (1 µM) for 6 days. For BET inhibitor/degrader exposure, cells were treated with fulvestrant plus palbociclib for 6 days then with respective BET inhibitor/degrader for 4 days. For TAM studies, cells were dosed with TAM (5 µM) for 4 days. For BET inhibitor/degrader exposure, cells were treated with TAM for 4 days then with respective BET inhibitor/degrader for 4 days. For validation of the knockout by CRISPER/Cas9, MCF-7 p53^−/−^ cells were treated with doxorubicin (D1515, Sigma-Aldrich; St. Louis, MO, USA) at either 1.0 µM or 3.0 µM, and cells were harvested after 24 h for Western blot analysis for p53 protein expression levels.

The experimental scheme for the current work was to evaluate the following: (i) Sensitization by ARV-825 to tamoxifen in MCF-7 p53 wild-type cells; (ii) Sensitization by ARV-825 to tamoxifen in MCF-7 p53^−RV^ cells. (iii) Sensitization by ABBV-744 to fulvestrant plus palbociclib in MCF-7 p53 wild-type cells. (iv and v) Sensitization by ABBV-744 to fulvestrant plus palbociclib in MCF-7 p53^−BB^ cells and T47D (p53 mutant) cells. (vi) Sensitization by ABBV-744 to TAM in MCF-7 p53 wild-type cells.

### 2.4. Cell Viability

The MTT and MTS assays were largely utilized as general screening protocols for drug action, but not for discrimination between growth arrest, growth inhibition, and cell death. For the MTS assay, cells were plated in a 96-well plate at a suitable concentration and were treated under the indicated conditions. The assay was performed according to the manufacturer’s protocol (AB197010, Abcam) and absorbance, which correlates to the number of viable cell per well, was recorded at 490 nm on a Biotek ELX800 Universal Microplate Reader. For the MTT assay, MCF-7 p53 WT and MCF-7 p53^−/−^ cells were seeded at a density of 2000 cells per well in 96-well and were treated under the indicated conditions; the MTT assay was performed according to the manufacturer’s protocol (CT01, Millipore Sigma). Subsequent to the solubilization of the purple formazan product (which correlates to the number of viable cells per well), plates were read on a TECAN Spectrafluor Plus with a test wavelength of 570 nm and a reference wavelength of 630 nm.

Trypan blue exclusion was utilized in the temporal response assays to distinguish between growth inhibition, growth arrest, and cell death. Cells were plated at 20,000 cells per well in a 6-well plate and treated with the respective agents. Cells were trypsinized, stained with 0.4% trypan blue (T01282, Sigma), and counted on the indicated days using a hemocytometer, and growth curves were generated from the collected data.

### 2.5. LDH Assay

The LDH assay was utilized for detection of cell death. MCF-7 cells were plated at 2000 cells per well in 96-well plates and treated with different concentrations of ABBV-744 (25, 50, 75, and 100 nM) for four days. The LDH assay was performed using CyQUANT LDH Cytotoxicity Assay Kit (C20301, Invitrogen) according to the manufacturer’s protocol and absorbance was recorded at 490 nm on a Biotek ELX800 Universal Microplate Reader to determine LDH release.

### 2.6. Promotion of Apoptosis

The extent of apoptotic cell death was measured using Annexin V-FITC/Propidium iodide staining. On the indicated day, cells were trypsinized, washed with 1X PBS and stained according to manufacturer’s protocol (556547, Annexin V-FITC Apoptosis Detection Kit; BD Biosciences; Franklin Lakes, NJ, USA). Fluorescence was quantified via flow cytometry using BD FACS Canto II and BD FACS Diva software at the Flow Cytometry Core Facility at Virginia Commonwealth University. For all flow cytometry experiments, 10,000 cells per replicate were analyzed, and three replicates for each condition were analyzed per independent experiment unless otherwise stated. All experiments were performed with cells protected from light.

### 2.7. Senescence-Associated β-Galactosidase (SA-ß-gal)

Cells treated as indicated were stained for β-galactosidase activity to determine the senescence phenotype according to the manufacturer’s protocol (ab65351, Abcam). Phase contrast images of cells were taken using a brightfield inverted microscope (20X objective, Q-Color3™ Olympus Camera; Olympus, Tokyo, Japan).

To quantify β-galactosidase-positive senescent cells, after the indicated treatment, cells were treated with Bafilomycin A1 (100 nM) for 1 h to achieve lysosomal alkalinization, followed by staining with C_12_FDG (10 μM) for 1 h at 37 °C. After incubation, cells were collected and analyzed using BD FACSCanto II and BD FACSDiva software. For all flow cytometry experiments, 10,000 cells per replicate were analyzed, and three replicates for each condition were analyzed per independent experiment unless otherwise stated. All experiments were performed with cells protected from light.

### 2.8. Western Blot Analysis

Western blotting was performed as previously described [17]. Briefly, after the indicated treatments, cells were trypsinized, harvested, and washed with 1X PBS. Pellets were lysed and protein concentrations were determined via the Bradford Assay (5000205, Bio-Rad Laboratories; Hercules, CA, USA). Protein samples were loaded and subjected to SDS-polyacrylamide gel electrophoresis, transferred to polyvinylidene difluoride membrane, and blocked with 5% BSA in 1X PBS with 0.1% Tween 20 (BP337, Fisher; Hampton, NH, USA). The membrane was incubated overnight at 4 °C with the indicated primary antibodies at a dilution of 1:1000 (except for c-Myc 1:250) with 5% BSA in 1X PBS. The membrane was then washed, secondary antibody was added at a dilution of 1:2000 with 5% BSA in 1X PBS for 2 h at room temperature, and the membrane was washed three times in 1X PBS with 0.1% Tween 20. Blots were developed using Pierce enhanced chemiluminescence reagents (32132, Thermo Scientific) on Bio-Rad ChemiDoc System. Image-J software was utilized for quantification of Western blots.

### 2.9. Immunoprecipitation

p53 (DO-1, Santa Cruz) or BRD4 (E2A7X, Cell Signaling Technology) primary antibodies (1:100 dilution) were added to equal amounts of whole cell lysates and incubated with rotation at 4 °C overnight. Antibody complexes were then captured using Protein A/G UltraLink Resin (53132, Thermo Fisher) at 4 °C with rotation for 1 h. Samples were centrifuged, washed three times with CHAPS buffer, and resuspended in CHAPS buffer and 5x SDS loading buffer. After boiling for 5 min, samples were analyzed via Western blotting as described above.

### 2.10. Statistics

Unless otherwise indicated, all quantitative data are shown as mean ± SEM from at least three independent experiments (biological replicates), all of which were performed in triplicates or duplicates (technical replicates). GraphPad Prism 9.0 software was used for statistical analysis. All data were analyzed using either a one- or two-way ANOVA, as appropriate, with Tukey or Sidak post hoc.

## 3. Results

### 3.1. ARV-825 Delays the Recovery of Tamoxifen-Treated MCF-7 Cells

Tamoxifen (TAM), one of the oldest and most frequently utilized SERMs, is now typically prescribed to treat hormone receptor-positive, early-stage breast cancer after surgery to reduce disease recurrence in pre-menopausal women [5,18]. There are emerging efforts [19,20] for potentially increasing sensitivity to TAM by the utilization of BET inhibitors in combination with TAM. Therefore, we investigated the influence of ARV-825 on the response to TAM in models of ER^+^ breast cancer.

Lee et al. [21] showed that TAM, at 10 µM, induced senescence in the MCF-7 cell line. However, treatment with a more clinically relevant concentration of TAM, 5 μM for 4 days, did not drive MCF-7 cells into a significant degree of senescence, as shown via β-galactosidase staining (Figure 1A) and flow cytometric quantification of C_12_FDG fluorescence, a metabolite for SA-ß-gal (Figure 1D). Instead, TAM treatment for 4 days resulted in a non-senescent transient growth suppression with ~20% apoptosis as compared to ~10% apoptosis in the control (Figure 1B,C).

ARV-825 utilized as a single agent also induced a transient growth arrest with a slight increase in the extent of apoptosis (Figure 1B,C), and without any detectable promotion of senescence (Figure 1D). Further, the addition of ARV-825 to cells treated with TAM caused an initial pronounced reduction in cell number without a significant increase in the extent of apoptosis (Figure 1B,C). Importantly, ARV-825-mediated inhibitory effects significantly suppressed and delayed the proliferative recovery (beginning at day 12) that is observed with TAM alone (beginning at day 6) (Figure 1B).

As would have been anticipated based on the established mechanisms of action of ARV-825 [8,22,23,24], ARV-825 either alone or in combination with TAM produced a significant reduction in BRD4 levels (Figure 1E) as well as suppressing the levels of BET family members, BRD2 and BRD3 (Appendix A). ARV-825 also collaterally produced a pronounced reduction in the BRD4 downstream effector, c-Myc, either alone or in combination with TAM (Figure 1E). These results are consistent with established mechanisms of action of ARV-825 in downregulating BRD proteins [8,22,23,24].

### 3.2. ARV-825 Effects in Combination with Tamoxifen Are Further Confirmed in the p53 Knockout MCF-7 Breast Tumor Cell Line

In our previous studies with fulvestrant plus palbociclib, ARV-825 enhanced the response in both the p53 wild-type (WT) MCF-7 cells as well as in the p53 mutant T47D cells [8]. In order to assess whether the p53 status of the cells would influence the response to ARV-825, we investigated the effect of ARV-825 on sensitivity to TAM in the p53^−/−^ MCF-7 ER^+^ breast cancer cell line where p53 had been silenced using CRISPR/Cas9. The p53 status of the cells was validated by Western blotting, which showed the absence of p53 even after the treatment with different concentrations of doxorubicin [25] (Figure 2A). In contrast, both p53 as well as its immediate downstream effector, p21, were clearly induced by doxorubicin in the MCF-7 p53 WT cells (Figure 2A).

Similarly to what was observed in MCF-7 p53 WT cells, ARV-825 alone inhibits cell growth (Figure 2B) without promotion of apoptosis (Figure 2C) or senescence (Figure 2D) in the MCF-7 p53^−/−^ cells. Furthermore, ARV-825 in combination with TAM suppressed and delayed proliferative recovery without promotion of either apoptosis or senescence (Figure 2B–D). Additionally, ARV-825 either alone or in combination with TAM suppressed both BRD4 as well as c-Myc in these cells (Figure 2E).

These results highlight the efficacy of ARV-825 in suppressing the proliferative capabilities of TAM-treated cells in a p53-independent manner, together with suppressing the levels of BRD4 as well as its downstream effector, c-Myc.

### 3.3. ABBV-744 Extends the Growth Inhibitory Effect Initiated by Fulvestrant plus Palbociclib in MCF-7 Cells

One major problem that is likely to limit the utilization of BET inhibitors such as AZD5153, BMS 986158, and CPI-0610 is the serious side effects [15]. These adverse effects range from mild symptoms, including diarrhea, nausea, and fatigue, to serious symptoms such as thrombocytopenia, anemia, as well as neutropenia. Consequently, we investigated another selective BET inhibitor with a more promising side effect profile, ABBV-744 [16]. ABBV-744 is a selective BDII domain inhibitor that has shown promising preclinical results in hematologic malignancies [26] as well as prostate cancer [16]. Therefore, we investigated the effect of ABBV-744 in various ER^+^ breast cancer models together with endocrine therapies.

We initially assessed the response of MCF-7 cells over a range of ABBV-744 concentrations (between 12.5 and 125 nM). ABBV-744 triggered moderate growth arrest at all concentrations tested (Appendix A), with minimal apoptosis (Appendix A). The lack of cell death was further confirmed via the LDH assay (Appendix A).

We next evaluated the effects of different ABBV-744 concentrations on the BET family members, BRD2, BRD3, and BRD4. ABBV-744 did not significantly alter the levels of BRD2 or BRD3 (Appendix A); however, a significant downregulation in BRD4 levels was evident at both 75 nM and 100 nM (Appendix A), indicating the potential selectivity of ABBV-744 towards BRD4 in ER^+^ breast cancer cells. Consequently, we chose 100 nM as an appropriate concentration for the subsequent studies and established that ABBV-744 (100 nM) as a single agent induces a modest but significant growth inhibitory response (Figure 3A), without significant apoptosis induction, as indicated by Annexin V/PI flow cytometry (Figure 3B).

Consistent with our previous study [8], we next confirmed that the combination of fulvestrant plus palbociclib drives cancer cells into a state of senescence, from which the cells escape between days 12 and 18 (Figure 3A,C). Importantly, the addition of ABBV-744 (100 nM) at days 6 to 10 extended the growth inhibitory state mediated by fulvestrant plus palbociclib, with cell recovery beginning at day 16 (Figure 3A). Furthermore, the extended growth arrest induced by ABBV-744 does not reflect an increase in the extent of senescence that was initiated by fulvestrant plus palbociclib, as shown via flow cytometry quantification of C_12_FDG fluorescence; that is, the extent of senescence was essentially identical for fulvestrant plus palbociclib alone and with the addition of ABBV-744 (Figure 3D).

ABBV-744 (100 nM) as a single agent or when used in combination with fulvestrant plus palbociclib causes a pronounced reduction in BRD4 levels (Figure 3E and Appendix A). In contrast, Appendix A shows that concentrations below 50 nM do not cause a reduction in BRD4 levels. The inability of 50 nM to reduce BRD4 levels corresponds with the observation that 50 nM ABBV-744 neither prolongs the growth arrest mediated by fulvestrant plus palbociclib, nor delays proliferative recovery (Appendix A), supporting the premise that BRD4 reduction is required for ABBV-744-mediated extended growth arrest.

Interestingly, ABBV-744 induces p53 accumulation in combination with fulvestrant plus palbociclib (Figure 3E), thus suggesting a possible role for the p53 in growth inhibition mediated by ABBV-744, in addition to BRD4. Therefore, we performed a co-immunoprecipitation assay (Figure 3F), which shows an association between p53 and BRD4 in the MCF-7 p53 WT cells (non-treated cells); specifically, BRD4 immunoprecipitation brings down p53, and conversely, p53 immunoprecipitation brings down BRD4.

### 3.4. ABBV-744 Fails to Extend the Growth Inhibitory Response Initiated by Fulvestrant plus Palbociclib in MCF-7 Cells Lacking p53

To investigate a putative role for p53 in mediating the effects of ABBV-744, we performed studies in MCF-7 p53^−/−^ cells. As shown for all of the cell models studied, fulvestrant plus palbociclib causes growth arrest in MCF-7 p53^−/−^ cells, with the cells beginning to recover between days 8 and 14 (Figure 4B). ABBV-744 alone only transiently and marginally suppressed MCF-7 p53^−/−^ breast tumor cell growth at 96 and 120 h; inhibitory effects were no longer evident at 144 h (Figure 4A). Furthermore, ABBV-744 failed to enhance or prolong the growth inhibitory effects of fulvestrant plus palbociclib or to interfere with proliferative recovery (Figure 4B). With respect to BRD4 and c-Myc, ABBV-744 alone or in combination with fulvestrant plus palbociclib suppressed the levels of BRD4 (Figure 4C), whereas c-Myc was barely detectable after fulvestrant plus palbociclib alone; furthermore, this suppression of c-Myc was not altered by the addition of ABBV-744 (Figure 4C). These studies confirm an apparent p53 dependency for sensitization to fulvestrant + palbociclib by ABBV-744, since there was no sensitization despite the pronounced effects on BRD4 levels.

These results, together with the data in Figure 3 and Appendix A, strongly suggest that ABBV-744 mediated growth inhibition of the cells treated with fulvestrant plus palbociclib is dependent on BRD4, as well as p53.

### 3.5. ABBV-744 Alone or in Combination with Fulvestrant plus Palbociclib Did Not Affect the Proliferation of T47D Cells

Approximately 20% of ER positive breast cancers present with p53 mutations [27]. Consequently, we further investigated whether ABBV-744 could also enhance the response to fulvestrant plus palbociclib in the p53 mutant T47D ER^+^ breast cancer cell line, as previously shown for ARV-825 [8]. Figure 5A–C demonstrate that ABBV-744, in the dose range between 25 and 100 nM, induces neither a growth inhibitory response, as shown by the MTS proliferation assay, nor apoptosis. Further, ABBV-744 (100 nM) does not extend the growth inhibitory state initiated by fulvestrant plus palbociclib in T47D cells (Figure 5D). The Annexin V/PI assay showed the lack of apoptosis either alone or in combination with fulvestrant plus palbociclib (Figure 5E).

ABBV-744 alone or in combination with fulvestrant plus palbociclib reduced BRD4 levels, while having little effect on c-Myc levels (Figure 5F and Appendix A), as was the case with the MCF-7 p53^−/−^ cells (Figure 4C), confirming the crucial role played by p53. These results are in stark contrast to the robust reduction in c-Myc levels induced by the combination of ARV-825 with fulvestrant plus palbociclib described previously [8] or in combination with TAM as shown previously (Figure 1E and Figure 2E). These studies indicate that the growth-suppressive effects mediated by ABBV-744 are correlated to BRD4 levels as well as p53 status.

### 3.6. ABBV-744 Does Not Extend the Growth Arrest Mediated by Tamoxifen in the MCF-7 p53 WT Cell Line

Although ABBV-744 was effective in enhancing the response to fulvestrant plus palbociclib in the MCF-7 p53 WT cells, Figure 6A shows that ABBV-744 does not prolong growth inhibition nor interfere with proliferative recovery in MCF-7 WT cells treated with TAM. There is also no increase in apoptosis over that observed with TAM alone (Figure 6B). A possible explanation for this lack of effect on MCF-7 p53 WT cell proliferation and recovery is likely related to the minimal suppression of BRD4 level when ABBV-744 is combined with TAM, despite an approximately 50% reduction in BRD4 using ABBV-744 alone (Figure 6C and Appendix A). Furthermore, showing consistency with our previous data, ABBV-744 alone or in combination with TAM did not affect the levels of c-Myc. Again, as was the case shown in Figure 3E, ABBV-744 induced p53 accumulation either alone or in combination with TAM (Figure 6C).

## 4. Discussion

Estrogen receptor alpha positive (ER^+^) breast cancer is the most common form of breast cancer and is predicted to comprise approximately 70% of all breast cancer cases [4]. The current standard of care is a combination of endocrine therapy with adjuvant therapy, reducing disease mortality by up to 40% [6]. However, disease recurrence, both local and distant, is a major limitation associated with these standard of care therapies [8,28,29] as well as contributing to most hormone receptor-positive breast cancer deaths. Frequently, this recurrence is associated with a latent cell population that can escape from the dormant state and become more aggressive in nature [30]. Different mechanisms of tumor dormancy have been studied, one of which is potentially senescence [30,31,32]. Senolytics, senostatics, and senomorphics have been investigated recently for the possible elimination or at least extended suppression of dormant tumor cell populations [33,34,35,36,37].

Endocrine therapies comprise different classes, among which are selective estrogen receptor degraders (SERDs), such as fulvestrant, which in combination with CDK4/6 inhibitors, including palbociclib, is one of the standard care therapies for metastatic ER-positive/Her2-negative breast cancer [5]. Fulvestrant acts by blocking the estrogen receptor and triggering receptor degradation [38], while palbociclib induces Rb-dependent growth arrest via CDK4/6 inhibition [39]. Both effects result in the inhibition of estrogen-mediated gene expression [38,39]. Clinically, the combination of fulvestrant and palbociclib was shown to extend progression-free survival in breast cancer patients from 4.6 to 11.2 months [8,9,10,11]. However, disease recurrence ultimately limits the efficacy of this combination therapy. In this and our previous study [8], we showed that fulvestrant plus palbociclib drives tumor cells into senescence, from which the cells begin to escape at days 12 to 18. Investigating the potential utilization of BET inhibitors/degraders as adjuvant therapy to fulvestrant plus palbociclib, we showed that ARV-825 extended the growth inhibitory effect mediated by fulvestrant plus palbociclib, with potential senolytic activity. This was shown in p53 wild-type, p53 mutant, as well as Rb-deleted cell lines, where the ARV-825-mediated effects were associated with and presumably mediated via the degradation of BRD4, and the suppression of its downstream effector, c-Myc [8]. These effects on c-Myc may prove to be clinically relevant given the recent publication by Mo et al. [40], which demonstrated that S6K, a downstream target of c-Myc, is involved in clinical expression of CDK4/6 resistance.

Another BET inhibitor, ABBV-744, that is being investigated in clinical trials for relapsed/refractory acute myeloid leukemia (AML) as well as myelofibrosis (NCT03360006, NCT04454658) has shown promising results in prostate cancer models [16]. In the current work, ABBV-744 extended the growth inhibitory response mediated by fulvestrant plus palbociclib at the concentration where BRD4 was downgraded without increasing either the magnitude of senescence or apoptosis in the MCF-7 cell population. ABBV-744 inhibits BRD4, which is apparently required for ABBV-744-mediated growth inhibition in combination with fulvestrant plus palbociclib. Interestingly, ABBV-744 appeared to demonstrate a p53 dependency, as indicated by studies in the p53 mutant T47D cell line as well as by CRISPR-mediated KO of p53 in MCF-7 cells. This p53 dependency is consistent with several publications [41,42,43,44] that have investigated the possible relationship between p53 and BET proteins, especially BRD4 [45]. Recently, Wu et al. [41] demonstrated that BRD4 interacts with p53 and showed that this interaction is modulated by casein kinase II (CK2). CK2 mediated the phosphorylation of a conserved acidic region in the BRD4 protein, dictating the chromatin binding of BRD4 as well as recruiting p53 to regulated promoters [41]. Based on the literature and our current findings, further investigation is necessary to understand the relation between BRD4 and p53 in order to validate the hypothesis that ABBV-744 may be disrupting the connection between p53 and BRD4, thereby affecting tumor cell growth.

Another class of endocrine therapies involves the utilization of selective estrogen receptor modulators (SERMS) including tamoxifen (TAM). TAM is one of the oldest and most frequently utilized SERMs, which competes with estrogen at the receptor site, blocking the promotion of breast cancer by estrogen [46]. TAM is typically prescribed to treat pre-menopausal women with early stages of hormone receptor-positive breast cancer after surgery to reduce disease recurrence [5].

In studies evaluating ABBV-744 in combination with TAM in MCF-7 cells, we observed that ABBV-744 did not induce either apoptosis or growth arrest. Furthermore, whereas BRD4 levels were reduced upon treatment with ABBV-744 alone, there was a barely detectable change in BRD4 level when ABBV-744 was combined with TAM. Showing consistency with our earlier data, there was no change in c-Myc levels when ABBV-744 was added either alone or in combination with TAM. In this context, several studies that have investigated the relationship between TAM and p53 have reported that p53 status is frequently associated with the reduced response to TAM and/or the development of TAM resistance [47,48,49,50,51,52,53,54]. Bailey et al. [55] have shown that TAM promotes antagonism to p53; Guillot et al. [56] reported that TAM treatment causes alterations in p53-mediated transcription; and Fernandez-Cuesta et al. [57] also showed that TAM-mediated effects relate to p53 status, suggesting that TAM has anti-p53 effects. These findings may provide a further explanation as to why ABBV-744 did not suppress the recovery of TAM-treated cells, which may be related to TAM anti-p53 effects. This is consistent with our data, in which, despite the induction of p53, there is no concomitant increase in p21 levels (pre-clinical results from our lab) when ABBV-744 is combined with TAM, indicative of an anti-p53 effect of TAM in this experimental model system.

In contrast to somewhat more limited sensitizing actions of ABBV-744, ARV-825 showed promising results in combination with TAM. ARV-825 suppressed the TAM-treated population in MCF-7 p53 WT cells, as well as in p53 KO cells [58]. Mechanistically, and as was the case with fulvestrant plus palbociclib [8], ARV-825 demonstrated a reduction in BRD2, BRD3, and, most prominently, BRD4, together with suppression of c-Myc, either alone or in a more pronounced fashion in combination with TAM. Therefore, the anti-proliferative activity of ARV-825 may be mediated through BRD4 inhibition/degradation [45], leading to the suppression of BRD4′s downstream effector, c-Myc, which is consistent with the findings of other investigators [22,23,24]. Notably, the current study also showed that ARV-825 is not acting solely as a senolytic, as TAM did not induce senescence at a clinically relevant concentration (5 µM).

Overall, our current work highlights the potential utilization of ARV-825 in combination with TAM. While ABBV-744 could also suppress proliferative recovery after fulvestrant plus palbociclib, its capacity to sensitize ER^+^ breast tumor cells was somewhat more limited. Additional studies in progress suggest that ARV-825 might also be effective in enhancing the response to estrogen deprivation (aromatase inhibition), another component of standard of care in ER^+^ breast cancer.

There are clearly limitations to the current work that need to be addressed in future studies. All of the experiments presented were performed in cell culture, and studies in tumor-bearing animals will be necessary to confirm that the proposed strategies are effective in vivo. It would further be critical to assess whether the addition of BET inhibitors, such as ARV-825 [59], exacerbates the toxicity of antiestrogen focused therapies. In the case of tamoxifen, the focus would be on hepatotoxicity, where alterations in the levels of enzymes such as alanine aminotransferase and aspartate aminotransferase would be monitored [60]. In the case of palbociclib, either alone or in combination with fulvestrant, the concern would be neutropenia [61,62,63], which could also be evaluated in an animal model system. In addition, studies utilizing ER^+^ breast tumor cells that have developed resistance to tamoxifen, fulvestrant, and/or palbociclib are needed to extend this work to the clinical situation, where drug resistance often compromises the effectiveness of antiestrogen therapies.

## Figures and Tables

**Figure 1 cancers-15-04066-f001:**
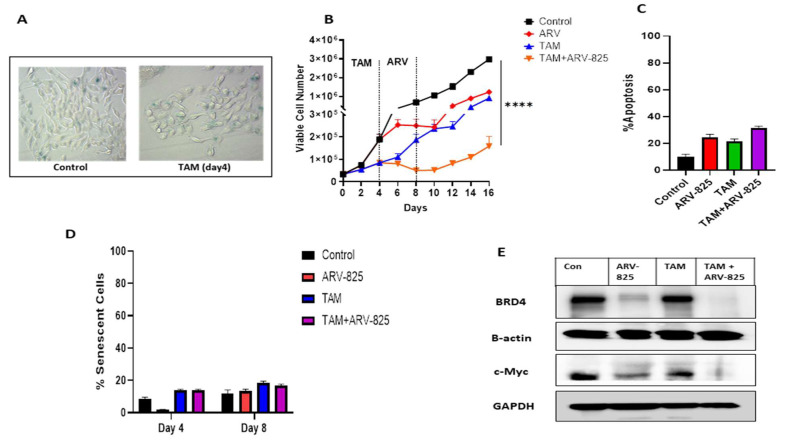
ARV-825 suppresses proliferative recovery of TAM-treated MCF-7 cells. (**A**) MCF-7 cells were treated with 5 µM TAM for 4 days. Cells were fixed on day 4, stained with x-gal staining solution, and imaged using a bright field microscope. All images were generated under the same magnification (20X). (**B**) Cells were treated with TAM (5 µM) for 4 days followed by ARV-825 (50 nM) addition for 4 days. Cell viability was monitored over a period of 16 days through trypan blue exclusion. (**C**) Cells were treated with TAM (5 µM) for 4 days followed by ARV-825 (50 nM) for 4 days. Apoptosis was evaluated at day 4 of ARV-825 treatment via flow cytometry using an Annexin V-FITC Apoptosis Detection Kit. (**D**) Percentage of SA-β-gal positive was quantified using C_12_FDG at the indicated time points. (**E**) Western blotting for BRD4 and c-Myc at day 4 of ARV-825 treatment. All images are representative fields or blots from at least two/three independent experiments. The uncropped blots are shown in the Appendix A. **** *p* ≤ 0.001 indicates statistical significance of each condition compared to the control as determined using two-way ANOVA with Sidak’s post hoc test.

**Figure 2 cancers-15-04066-f002:**
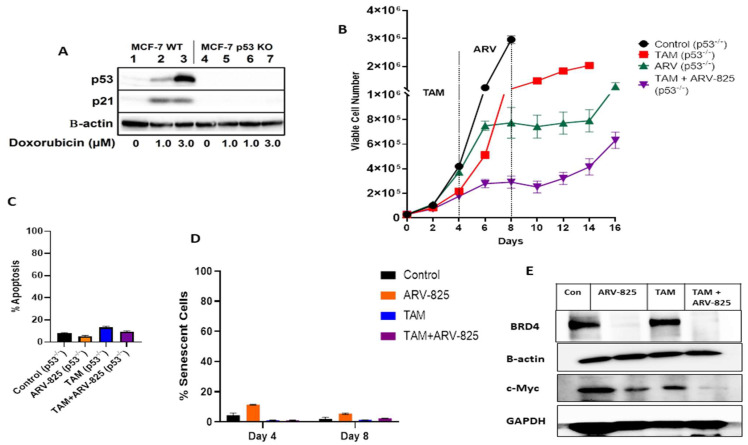
ARV-825 induces significant growth suppression in combination with TAM in MCF-7 p53^−/−^ cells. (**A**) Western blot for p53, and p21 after 24 h of doxorubicin treatment. (**B**) MCF-7 p53^−/−^ cells were treated with TAM (5 µM) for 4 days followed by ARV-825 (50 nM) addition for 4 days. Viable cell number was monitored over a period of 16 days through trypan blue exclusion. (**C**) Apoptosis was evaluated via flow cytometry using an Annexin V-FITC Apoptosis Detection Kit. (**D**) Percentage of SA-β-gal positive was quantified using C_12_FDG at the indicated time points. (**E**) Western blotting for BRD4 and c-Myc at day 4 of ARV-825 treatment. All images are representative fields or blots from at least two/three independent experiments. The uncropped blots are shown in the Appendix A.

**Figure 3 cancers-15-04066-f003:**
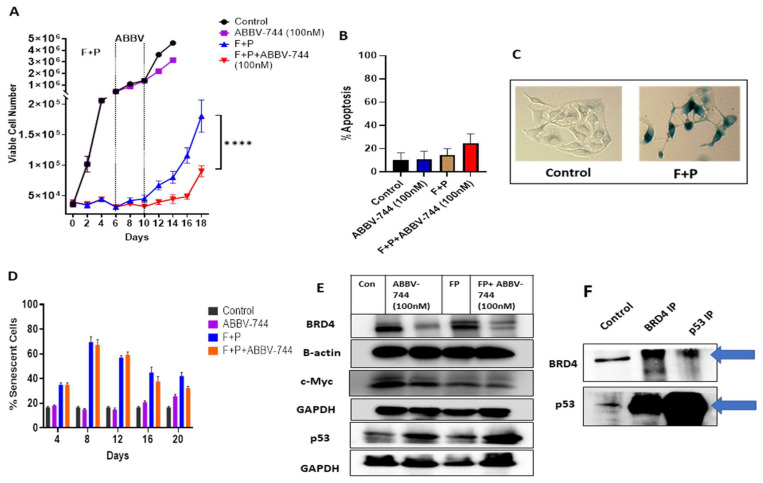
ABBV-744 extends the growth inhibitory response initiated by fulvestrant plus palbociclib in MCF-7 cells. (**A**) Cells were treated with fulvestrant (100 nM) plus palbociclib (1 µM) for 6 days followed by ABBV-744 (100 nM) addition starting from day 6 to 10. Cell viability was monitored over a period of 18 days through trypan blue exclusion. (**B**) Cells were treated with fulvestrant (100 nM) plus palbociclib (1 µM) for 6 days followed by ABBV-744 (100 nM) from day 6 to 10. Apoptosis was evaluated at day 4 of ABBV-744 treatment via flow cytometry using an Annexin V-FITC Apoptosis Detection Kit. (**C**) Cells were treated with fulvestrant (100 nM) plus palbociclib (1 µM) for 6 days and fixed on Day 6, stained with x-gal staining solution, and imaged using a bright field microscope. Both images were generated under the same magnification (20X). (**D**) Quantification of SA-β-gal using C_12_FDG at the indicated time points. (**E**) Western blotting for BRD4, c-Myc, and p53 at day 4 of ABBV-744 treatment. (**F**) MCF-7 p53 WT cells were subjected to immunoprecipitation for BRD4 and p53 using BRD4 and p53 antibodies. The arrows indicate the location of BRD4 and p53 bands at 152 kDa and 53 kDa, respectively. All images are representative fields or blots from at least two/three independent experiments. The uncropped blots are shown in the Appendix A. **** *p* ≤ 0.001 indicates statistical significance of each condition compared to fulvestrant plus palbociclib as determined using two-way ANOVA with Sidak’s post hoc test.

**Figure 4 cancers-15-04066-f004:**
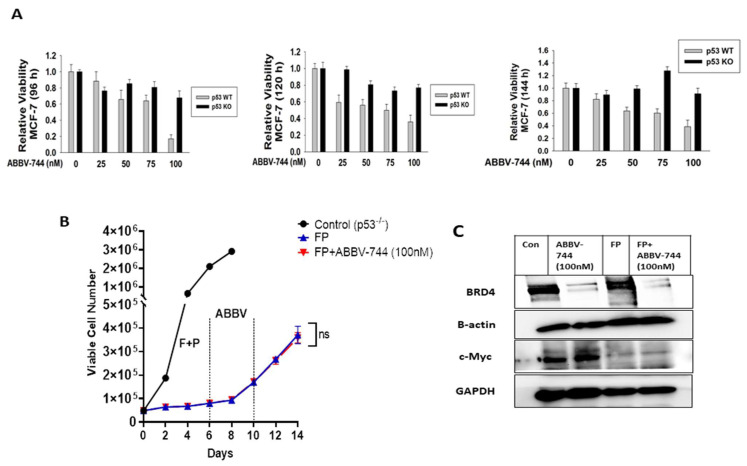
Absence of ABBV-744 effects in MCF-7 p53^−/−^ cells. (**A**) Both MCF-7 p53 WT and p53^−/−^ cells were treated with different concentrations of ABBV-744 (25 nM, 50 nM, 75 nM, and 100 nM) for 4 days; cell viability was determined after 96, 120, and 144 h using the MTT assay. (**B**) MCF-7 p53^−/−^ cells were treated with fulvestrant (100 nM) plus palbociclib (1 µM) for 6 days followed by ABBV-744 (100 nM) addition starting from day 6 to 10. Cell viability was monitored over a period of 14 days through trypan blue exclusion. (**C**) Western blotting for BRD4 and c-Myc at day 4 of ABBV-744 treatment. All images are representative fields or blots from at least two/three independent experiments. The uncropped blots are shown in the Appendix A. NS indicates non-statistical significance of each condition compared to fulvestrant plus palbociclib as determined using two-way ANOVA with Sidak’s post hoc test.

**Figure 5 cancers-15-04066-f005:**
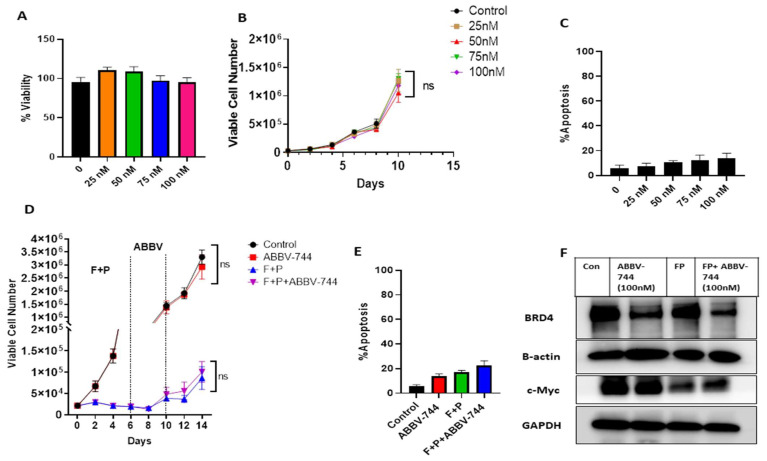
ABBV-744 did not extend growth arrest when combined with fulvestrant plus palbociclib in T47D cells. (**A**) Cells were treated with ABBV-744 (25 nM, 50 nM, 75 nM, and 100 nM) for 4 days. Percent cell viability was measured by the MTS viability assay. (**B**) Cell viability was monitored over a period of 10 days through trypan blue exclusion. (**C**) Apoptosis was evaluated via flow cytometry using an APC Annexin V Apoptosis Detection Kit. (**D**) Cells were treated with fulvestrant (100 nM) plus palbociclib (1 µM) for 6 days followed by ABBV-744 (100 nM) addition for 4 days. Cell viability was monitored over a period of 14 days through trypan blue exclusion. (**E**) Apoptosis was evaluated via flow cytometry using an Annexin V-FITC Apoptosis Detection Kit. (**F**) Western blotting for BRD4, and c-Myc at day 4 of ABBV-744 treatment. All images are representative fields or blots from at least two/three independent experiments. The uncropped blots are shown in the Appendix A. NS indicates non-statistical significance of each condition compared to control and fulvestrant plus palbociclib as determined using two-way ANOVA with Sidak’s post hoc test.

**Figure 6 cancers-15-04066-f006:**
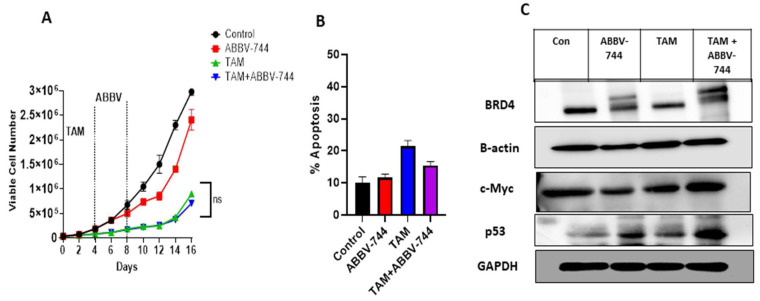
ABBV-744 did not suppress the recovery of TAM-treated cells. (**A**) Cells were treated with TAM (5 µM) for 4 days followed by ABBV-744 (100 nM) addition starting from day 4 to day 8. Cell viability was monitored over a period of 16 days through trypan blue exclusion. (**B**) Apoptosis was evaluated via flow cytometry using an Annexin V-FITC Apoptosis Detection Kit. (**C**) Western blotting for BRD4, c-Myc, and p53 at day 4 of ABBV-744 treatment. All images are representative fields or blots from at least two/three independent experiments. The uncropped blots are shown in the Appendix A. NS (non-significant) indicates non-statistical significance of TAM plus ABBV-744 compared to TAM alone as determined using two-way ANOVA with Sidak’s post hoc test.

## Data Availability

The original contributions presented in the study are included in the article/Appendix A. Further inquiries can be directed to the corresponding authors.

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
