# Peer review of "BRD4 Inhibition as a Strategy to Prolong the Response to Standard of Care in Estrogen Receptor-Positive Breast Cancer"

_cancers, 2023, doi:10.3390/cancers15164066_

Round 1

Reviewer 1 Report

Ahmed et al. present a research article demonstrating the improvement of the response to CDK4/6 inhibitors plus fulvestrant or tamoxifen by BRD4 inhibiors in ER positive breast cancer model. The data showed that ARV-825 was effective in both p53 wild-type (WT) and in p53-/- cells either alone or in combination with tamoxifen, whereas the effectiveness of ABBV-744 was limited to fulvestrant + palbociclib in p53 WT cells. The differential effects may be correlated to BRD4-mediated regulation of c-Myc. Overall, the study showed good data quality, and the manuscript is prepared in a proper format. The weakness of this study is the mechanism of drug combination-mediated suppression of cell growth, since neither apoptosis nor senescence were increased. The comments for the authors are listed as follows:

1.     There are some typos, such as ER+ and p53-/-, which should be superscript.  

2.     Why did the author use different methods to measure cell proliferation and viability? Such as trypan blue exclusion, MTS assay, and MTT assay, and the authors have to explain in the manuscript.

3.     The authors did not provide original (uncropped) gel blot of Western blotting, and the way to present the results can be improved.

4.     In Fig. 1C, please mention the treatment time in the legend.

5.     In Fig. 1E and Fig. 2E, ARV-825 alone depleted the expression of BRD4, and the combination of TAM did not further suppress its expression. How to explain the combination-mediated delayed of the recovery of tamoxifen-treated cells?

6.     Why did the author not choose palbociclib+Fulvestrant for the comnination with ARV-825?

7.     In Fig. 3B, please mention the treatment time in the legend.

8.     The authors concluded that ABBV-744-mediated growth inhibition of the cells treated with fulvestrant plus palbociclib is dependent on BRD4, as well as p53. But in Fig.2E, how to explain that ARV-825 was able to suppress BRD4 and c-Myc in p53 null MCF7 cells? What is the mechanism by which the interaction of BRD4 and p53 could regulate the expression of c-Myc?

9.     In Fig. 6A, the data of TAM alone already can inhibit the proliferation of MCF-7 cells, which is inconsistent with the data in in Fig. 1B and Fig. 2B where TAM alone rebound in 8 to 10 days of treatment.

10.  In this study, neither TAM alone nor fulvestrant plus palbociclib increased the expression of BRD4. What is the mechanism of drug combination-mediated improvement of cell-growth suppression?

There are some typos, such as ER+ and p53-/-, which should be superscript.  

Reviewer 2 Report

We have previously shown that overexpression of the epigenetic regulator BRD4 is a negative prognostic indicator in breast cancer, and BET family inhibitors such as ARV-825 and ABBV-744 have generated interest due to their ability to improve and prolong response to current therapeutic strategies. . Current work has explored the potential of using ARV-825 and ABBV-744 to improve the efficacy of tamoxifen or fulvestrant in combination with palbociclib. ARV-825 was effective in both wild-type (WT) p53 breast tumor cells and cells lacking functional p53 alone or in combination with tamoxifen, while ABBV-744 was limited to fulvestrant + palbociclib in p53 WT cells . The authors attribute these effects to the ability to suppress c-Myc, the downstream target of BRD4.

In general, I liked the article, but there are a few small comments:

1. It is necessary to give a general scheme of the experiment in the Materials and Methods section, since in the Results section information is given in parts (for individual cell lines), this will help to better perceive the material.

2. Pictures are unreadable in places. So, for example, in fig. 1B and D there are very small inscriptions of the legend; in Figs. C and E, you can zoom in along the y-axis, then the figures will look better and the comparison between groups will be more visual. The same applies to all other drawings.

Reviewer 3 Report

The manuscript by Elshazly, et al. describes the utility of inhibiting BRD4 to decrease recurrence of ER+ positive breast tumors, following treatment with estrogen antagonist and CDK4/6 inhibitors. The work presented used in vitro models to demonstrate that BET inhibitors (AV-825 and ABBV-744) have differential effects, depending upon the p53 status, to prolong growth arrest of ER+ breast cancer cell lines, following treatment with tamoxifen and palbociclib. Overall, the data supports the hypothesis and the mechanistic delineation of the inhibitor properties of AV-825 and ABBV-744. However, the utility of this finding in a clinical setting is not clear. What will be the sequential treatment strategy as tamoxifen resistant tumors can take many years to develop in a clinical setting.   

Major concerns:

1) The authors need to study the effects of this combination treatment on normal mammary epithelial cells.

2) In vivo studies should be presented.

3) The ability to overcome growth arrest using AV-825 and ABBV-744 should be performed using a clinically relevant drug resistant model.

4) Treatment strategy and protocol, in a clinical setting, should be discussed.

Round 2

Reviewer 1 Report

Thanks to the author's efforts. I have no more questions.

Author Response

Thank you for review our manuscript for publication in Cancers.

Reviewer 3 Report

Animal experiments would have significantly strengthened the conclusion. However, it is understandable that the time frame to complete these experiments would be long. With respect to the effect of the drug combination on normal mammary epithelial cells, this should be done and can be achieved in a relative short period of time.

Author Response

Thank you for the review our manuscript for publication in Cancers.

The primary dose-limiting toxicity of tamoxifen appears to be hepatotoxicity, while that of palbociclib, either alone or in combination  with fulvestrant, is neutropenia. It would therefore be of importance to determine whether these are exacerbated by the addition of the ARV-825, but in an animal model system rather than in cells in culture. We could modify the Discussion to reflect this as a future approach, as indicated below. 

here are clearly limitations to the current work that need to be addressed in future studies. All of the experiments presented were performed in cell culture, and studies in tumor-bearing animals will be necessary to confirm that the proposed strategies are effective in vivo. It would further be critical to assess whether the addition of BET inhibitors, such as ARV-825 (PMID: 34733788), exacerbates the toxicity of antiestrogen focused therapies. In the case of tamoxifen, the focus would be on hepatotoxicity, where alterations in the levels of enzymes such as alanine aminotransferase and aspartate aminotransferase would be monitored (PMID 26870344 ). In the case of palbociclib, either alone or in combination with fulvestrant, the concern would be neutropenia ( PMID  31391852;  PMID: 30895534; PMID 30136059 ), which could also be evaluated in an animal model system.

Round 3

Reviewer 3 Report

Generally, limitations of the data presented in the manuscript are included when the overall data presented is supporting the conclusions derived from the experimental data. The data presented in this manuscripts lacks the scientific rigor to derive the conclusions drawn. At a minimum, performing the in vitro work will support and strengthen the overall conclusion.
